# Structure, Thermal Properties and Proton Conductivity of the Sulfonated Polyphenylquinoxalines

**DOI:** 10.3390/membranes12111095

**Published:** 2022-11-03

**Authors:** Anna V. Pisareva, Nataliya M. Belomoina, Elena G. Bulycheva, Mikhail M. Ilyin, Evgeniya Y. Postnova, Rostislav V. Pisarev, Tatiana S. Zyubina, Alexander S. Zyubin, Alexander I. Karelin, Yury A. Dobrovolsky

**Affiliations:** 1Federal Research Center of Problem of Chemical Physics and Medicinal Chemistry RAS, 142432 Chernogolovka, Russia; 2Nesmeyanov Institute of Organoelement Compounds RAS, 119334 Moscow, Russia; 3Osipyan Institute of Solid State Physics RAS, 142432 Chernogolovka, Russia

**Keywords:** sulfonated polyphenylquinoxalines, structure, quantum chemical modeling, thermal stability, degree of sulfonation, conductivity

## Abstract

This paper briefly reviews the results of scientific research on the proton conductivity of sulfonated polyphenylquinoxalines. Synthesis, structure (IR spectroscopy, SEM, quantum-chemical modeling, molecular weight distribution), moisture capacity, thermal properties, and proton conductivity of sulfonated polyphenylquinoxalines (sulfur content 2.6, 4.2, 5.5, and 7%) were studied. The relative stable configurations of sulfonated polyphenylquinoxalines with different positions of benzene rings and sulfogroups with the help of quantum chemical modeling were modeled. Sulfonation of the starting polyphenylquinoxalines was confirmed by IR spectroscopy and elemental analysis. The SEM method was used to study the surface of sulfonated polyphenylquinoxalines, and sulfonation regions were found. It was shown that sulfonated polyphenylquinoxalines contain water and are stable up to 250 °C; on further heating, the decomposition of the sulfogroups occurs. The conductivity of the obtained polymer electrolytes was studied by impedance spectroscopy, and long-term tests were carried out. It is shown that the proton conductivity at an ambient humidity of 98 rel. % reaches values 10^−6^–10^−3^ S/cm depending on the degree of sulfonation. It was shown that even after long-term storage in air (7 years), samples of sulfonated polyphenylquinoxalines with a high sulfur content of 7% at 98% air humidity have a conductivity of 8 × 10^−4^ S/cm.

## 1. Introduction

The synthesis of new polymer proton-conducting electrolytes [1], which are one of the main components of fuel cells [2,3], attracts considerable attention from researchers [4]. Especially intensive work is underway to create proton-conducting electrolytes based on sulfonated aromatic condensation polymers, which can act as an alternative to commercial membranes of the Nafion type: different aromatic condensation polymers, poly(arylene oxides), polybenzimidazole, sulfonated PEEK [1,2,3,4,5,6,7,8].

Among aromatic condensation polymers used to prepare polymeric proton-conducting membranes, there are few works devoted to polyphenylquinoxalines (PPQs), despite the fact that PPQs meet such important requirements as high thermal [9,10] and oxidative stability [11], high glass transition temperature, and good film-forming properties, due to solubility in organic solvents and the ease of polymer-analogous transformations. In particular, during sulfonation by various methods [12,13,14], one can obtain thermoreactive polymers [15], polymers in salts form [16], and modified polymers [17]. By themselves, PPQs are high-temperature polymer dielectrics due to their high thermal stability at elevated temperatures, good dielectric properties in a wide range of temperatures and frequencies, good mechanical properties, high environmental stability, etc. [18]. There are also data in the literature on doping PPQ with phosphoric acid for use in high-temperature fuel cells. Thus, in [19], a copolymer of PPQ and polybenzimidazole doped with phosphoric acid (58 mol. % PPQ—42 mol. % PBI—39.2 mol.% H_3_PO_4_) with a conductivity of 0.24 S/cm at 180 °C were obtained. In [20], a study of the effect of PPQ doping with phosphoric acid was carried out, and it was shown that the obtained proton-exchange membranes have a conductivity of 3 × 10^−3^ S/cm at 160 °C. The disadvantage of this type of membrane is the solubility of the original polymer in adsorbed phosphoric acid. An attempt to solve this problem was presented in [21], where the PPQ units were additionally crosslinked with sulfonic bridges, after which the mechanical properties were improved upon doping with phosphoric acid.

In the published works devoted to the electrochemical characteristics of membranes based on sulfonated PPQ (SPPQ), there are no systematic studies of the dependence of the proton conductivity of these membranes on the degree of sulfonation, temperature, and humidity. Thus, in [22], it was shown that the specific conductivity of SPPQ in electrolyte solutions could reach 10^−5^–10^−2^ S/cm.

The authors of [12] sulfonated the PPQ film with 50% H_2_SO_4_ for 2 h and heated it to 300 °C for 90 s. The conductivity of SPPQ with a moisture capacity of 51% and a content of 2.8 H_2_SO_4_ per monomer at room temperature was 9.8 × 10^−2^ S/cm. In [23], the SPPQ conductivity (91%) at 20 °C and 100% RH was 9.8 × 10^−3^ (8 water molecules per sulfogroup) and 1.3 × 10^−1^ S/cm at t_max_ = 180 °C. The conductivity of SPPQ (% SO_3_H 107) in a cell with water is 5 × 10^−3^ S/cm, t_max_ = 153 °C [24]. The SPPQ conductivity was measured in water and reached 0.1 S/cm; the swelling was 7.3% at 100 °C [25].

A study of the conductivity of SPPQ with a sulfur content of 0.7, 1.7, 6, and 8% was presented in [26,27]. It was shown that the conductivity of SPPQ 8% S increases to 10^−3^ S/cm at a humidity close to 100 rel. %. The membrane was tested as part of a hydrogen-air fuel cell; the maximum power of the membraneelectrode unit with the E-TEK catalyst ~80 mW/cm^2^.

The conducting properties of PPQ doped with an aqueous solution of sulfuric acid were studied [28]. The reversibility of doping was proved, and a study of the change in the color of the films on the concentration of sulfuric acid was presented. It was shown that conductivity 4.5 × 10^−2^ S/cm was achieved with doping 9.1 MH_2_SO_4_. Measurements after 3 months showed a decrease in conductivity within the order of value.

Side-chain type sulfonated poly(phenylquinoxaline) (SPPQ)-based proton exchange membranes (PEMs) with different ionic exchange capacities were synthesized in [29]. In the single-cell test, the maximum power density of side-chain type SPPQ-5 was 63.8 mW cm^−2^ at 20 wt% methanol solution and O_2_ at 60 °C. The SPPQ PEMs showed high performance (62.8 mW/cm^2^) even when the methanol concentration was as high as 30 wt%. The conductivity of the samples is 0.02–0.07 S/cm in the temperature range of 30–80 °C, and the energy of activation was 0.11–0.12 eV.

Conductive PPQ films were obtained with a conductivity of 10^−7^–10^−12^ S/cm at a humidity of 50 rel. % depending on the anion of the dopant (ClO_4_^−^, AsF_6_^−^, BF_4_^−^, PF_6_^−^), with an increase in humidity, the conductivity can reach 10^−4^ S/cm [30].It was shown that conducting PPQs contain water, which solvates dopant ions, are sensitive to changes in humidity, and the color of the films changes reversibly during hydration–dehydration processes.

Earlier in [31], UV spectroscopy was used to study the protonation reaction of PPQ in solutions of some acids (HF, HCl, HClO_4_, H_2_SO_4_, CH_3_SO_3_H, and NH_4_F). Three different states were found concerning completely reversible reactions.Each state has its own color: pale yellow, similarto that of PPQ; golden yellow; and purple-red for the first and second states of protonation.By using IR spectroscopy, it was shown that hydrogen atoms are fixed on the nitrogen atoms of the polymer.

The authors [32] investigated the acid-base properties of PPQ films in various concentrated acids. The work in [33] describes complexes of PPQ with strong protic acids. It was shown that although the conductivity of PPQ-acid films is rather high (10^−4^–10^−2^ S/cm), the loss of the acid complex in weakly acidic or neutral solutions or volatilization of the acid into the environment leads to an almost dielectric (<10^−7^ S /cm).The pK (a) values for PPQ.H-4 (4+) and PPQ.H-2 (2+) complexes determined by the spectroscopic method were −6.1 and −2, indicating that the acid-base interaction of PPQ and acid is very weak. Electrochemical oxidation and reduction in PPQ-acid complexes are reversible in solutions of strong acids. The charge density of the electrode film of protonated PPQ is 125 mAh cm^−3^ between −0.1 and 0.1 V.

The authors of [34] believed that the deformation of polymers had a very strong correlation with entanglement density. Entanglement is an important feature of polymers, which is one of the key factors controlling the rheological, viscoelastic, solid mechanical, and adhesive properties of polymers. For polymers with low chain entanglement density, the deformation mechanism is mainly crazing; for polymers with high entanglement density, the deformation mechanism is mainly shear yielding. For polymers with middle entanglement density, both crazing and shear yielding may be the case. In the works of the authors of [35,36,37], the effect of physical aging on mechanical properties was studied, and the mechanism of deformation of PPQ films was investigated. It was found by TEM that deformation occurs due to the cracking of fibers, consisting of microfibers and microvoids, and/or stretching. One of the ways to solve this problem of cracking is to obtain block copolymers. The authors of [38] obtained heat-resistant film materials with high mechanical characteristics based on rigid-chain polynaphthoylenimidobenzimidazole and flexible-chain PPQ. It was shown that copolymers are characterized not only by an excess of tensile strength by additive values but also by a significant (2–4) times increase in their deformability in relation to homopolymers.

In this work, the moisture capacity, thermal properties, and proton conductivity of sulfonated polyphenylquinoxalines (sulfur content 2.6, 4.2, 5.5, and 7%) in the powder state and in the form of films were considered. The synthesis of PPQ was carried out on the basis of 3,3’, 4,4´-tetraaminodiphenyloxide (TADO), and 1,4-bis(phenylglyoxalyl)benzene (BPGB); sulfonation of this polymer was also carried out, and SPPQ films for studies were obtained. By using quantum chemical modeling, the relative stable configurations of PPQs with different positions of benzene rings and sulfogroups and their manifestation in IR spectra were considered.

## 2. Materials and Methods

*Synthesis of the SPPQ.* The synthesis of PPQ was carried out by the interaction of equimolar amounts of 3,3′, 4,4′-tetraaminodiphenyloxide (TADO) and 1,4-bis (phenylglyoxalyl) benzene (BFGB) (Aldrich) according to [9,15,39] (Figure 1).

The synthesis of PPQ carried out by the interaction of TADO with BFGB in chloroform with the use of methanol as a proton donor led to the preparation of a high molecular weight polymer (η_graft_ = 0.65 dL/g, Mw = 33,274) completely soluble in chloroform, benzyl alcohol, N-MP, *m*-cresol, and H_2_SO_4_. Sulfonation of PPQ was carried out by heating the initial PPQ in a mixture of conc. H_2_SO_4_ with oleum (3:1) at 120 °C for 5–35h. The sulfonation degrees (sulfur content in PPQ) increased during the sulfonation process. The transformation of PPQ into SPPQ was accompanied by a significant change in solubility. Thus, in contrast to PPQ, SPPQ are insoluble in chloroform [12]. An increase in the softening temperature of polymers with an increase in the degree of sulfonation, noted by us earlier [12], is associated with the development of a system of intermolecular hydrogen bonds as the sulfonation process deepens [30].

In order to prepare films based on PPQ and SPPQ, the corresponding polymer was dissolved in NMP (the concentration of the solid phase was about 5–6%), after which the solution was poured onto a horizontal glass plate, heated at a temperature of 95 °C in an argon atmosphere for 24 h, then the resulting the film was dried under vacuum at 80 °C to constant weight. The thickness of the films obtained was about 50 μm. Figure 1 shows a photograph of the appearance of films with different degrees of sulfonation.

*Elemental analysis*. Determination of the sulfur content in polymer electrolytes was carried out on a universal elemental CNS analyzer Vario Micro Cube (Langenselbold, Germany). According to the elemental analysis data, the sulfur content in the SPPQ samples is 2.6, 4.2, 5.5, and 7%.

*Thermal stability*. The thermal properties of PPQ and SPPQ polymer electrolytes with a sulfur content of 2.6, 4.2, 5.5, and 7% were investigated in the temperature range of 25–300 °C by synchronous thermal analysis in an argon atmosphere; the heating rate was 5 °C/min using a device manufactured by the company Netzsch STA 409 PC Luxx^®^ (Selb, Bavaria, Germany).

*Water sorption properties*. The water content in the samples was determined by the gravimetric method, keeping an accurately weighed and dried weighed portion of PPQ and SPPQ at certain air humidity to constant weight. The equilibration time varied from one week to a month. In order to create the required ambient air humidity, the samples were kept over saturated solutions of inorganic salts in the humidity range of 10–98 rel. %.

*Determination of molecular weight.* Polymer solutions were prepared in NMP with the addition of triethylamine (not all samples were dissolved in pure NMP). The sulfur content in samples was 7, 4.2, and 0%. The study of the characteristics of individual macromolecules, as well as the molecular weight distribution, was carried out by gel permeation chromatography (GPC) on a Knauer liquid chromatograph (Berlin, Germany) with a UV detector (wavelength 254 nm) and column thermostatted at 25 °C [40]. A single Phenomenex–Phenogel column (300 × 7.8 mm, 5 μm) 10 ×10^4^ Å was used, which is based on a styrene-divinylbenzene crosslinked polymer gel. For this column, the working range of the determined molecular weights is in the range of 5000–500,000 Da. N-MP was used as a solvent for the studied polymers, as well as an eluent for GPC. The eluent flow rate was 1 mL/min. The column was pre-calibrated against polystyrene standards of known molecular weight. Data acquisition and subsequent calculation of all polymer characteristics were carried out using the Ekochrome software and hardware complex (Moscow, Russia).

*Scanning electronic microscopy.* High-resolution scanning electron microscope Supra 50VP (Zeiss, Munich, Germany) with INCA Energy + microanalysis system (Oxford, UK) was used.

*IR spectroscopy.* IR Absorbance and ATR IR spectra were recorded in the range 50–4000 cm^−1^ (50 to 100 scans, resolution 4 cm^−1^) on the spectrometer Bruker Vertex 70V FTIR (Bremen, Germany).

*Quantum chemical simulation*. The systems under study were modeled within the framework of the cluster approach using the hybrid density functional B3LYP [41,42] with valence-two-exponential basis 6–31G*, including polarization functions, using the software package GAUSSIAN (Gaussian Inc., Wallingford, CT, USA) [43].

*Impedance spectroscopy.* The proton transfer parameters of polymer electrolyte samples were determined by impedance spectroscopy. For measurements, we used symmetric cells Pt/test electrolyte/Pt (electrolyte powders pressed in the form of a tablet) and C/test electrolyte/C (electrolyte polymer film), kept at a certain value of ambient humidity in the range of 10–98 rel. %. The time to establish equilibrium with the environment was from one week to a month. The control of the establishment of equilibrium with the environment was carried out to stabilize the resistance of the measured cell for several days. The frequency dependences of the resistance of the samples were obtained on an impedancemeter Z-350M (LLC “Elins”, Chernogolovka, Russia) in the frequency range of 1 Hz–1 MHz. The frequency dependence of the resistance was analyzed by the graphical analytical method [44]. The volumetric resistance was determined by cyclic extrapolation of the impedance hodograph to the axis of active resistance at ω→*∞* (ω = 2πν—circular frequency). The temperature dependence in the range of 293–323 K of conductivity was approximated using the Arrhenius–Frenkel equation: σ*_i_T* = Aexp(−*E*/*kT*).

## 3. Results

### 3.1. Thermal Analysis

According to the STA data, SPPQ films with a sulfur content of 2.6, 4.2, 5.5, and 7% dried at a moisture content of 20 rel. % containing 5–7 wt. % water (Table 1) and were stable up to 250 °C; further heating was accompanied by the release of sulfur oxides SO_2_ and SO, and the initial PPQ did not decompose up to 500 °C (Figure 2).

Figure 3 shows the results of the decomposition of the SPPQ 5.5 film with a mass spectrometric analysis of the decomposition products. As can be seen from the figure, in the region up to 150 °C, decomposition occurs with the release of water; the decomposition of sulfogroups began when heated above 300 °C.

### 3.2. Water Sorption

The study of the influence of ambient humidity on the moisture absorption of the studied SPPQ samples showed that with an increase in humidity up to 98 rel.% the amount of adsorbed water increases slightly to 0.16–0.33 wt.%, in the case of SPPQ 7, up to 0.5 wt.% (Figure 4).

### 3.3. Molecular Weight

The molecular weight distribution in solutions of polymers PPQ and SPPQ in N-MP with the addition of triethylamine presents in Figure 5. The sulfur content in the samples was 0, 4, and 7%. As can be seen from the data presented, the samples of PPQ and SPPQ were characterized by significant polydispersity (Table 2), which appeared at the stage of the polycondensation process of obtaining PPQ and SPPQ during the sulfonation reaction. We determined the apparent molecular weights of a number of polymers with the same main chain and a different number of sulfogroups. The results obtained explained the initial increase in molecular weights during sulfonation, as sulfonated polymers have larger radii of gyration in solution due to electric repulsion among the sulfonic acid groups attached to the polymers. The subsequent decrease in molecular weights can be explained by the possible destruction with prolonged exposure to sulfonating agents. It should be noted that all synthesized polymers are film-forming and, therefore, sufficiently high-molecular.

### 3.4. Scanning Electronic Microscopy

As shown in [15], the deformation and strength characteristics of PPQ are higher than that of SPPQ; it was shown by AFM that the production of films from solutions with a low polymer concentration leads to the formation of an island structure; at a higher concentration, uniform, practically defect-free continuous films can be formed. Unfortunately, one of the drawbacks that hinder the practical use of the obtained SPPQ is the brittleness of the films under study. Figure 6a,b shows micrographs confirming the presence of cracks. Figure 6c,d shows the sulfonation regions of SPPQ. Figure 7 shows the analysis data on the sulfur content in different parts of the film SPPQ.

### 3.5. IR Spectroscopy

Figure 8 shows the survey IR absorption spectra of PPQ and SPPQ films, in which bands were found to be common for many aromatic compounds. It should be noted that the delocalization of most vibrational transitions is a characteristic feature of the simplest aromatic cycles. These vibrations cannot be attributed to individual bonds and (or) bond angles, with the exception of stretching vibrations of CH and bending vibrations of CCC. The unification of the cycles in the case of the formation of the PPQ molecule only complicates the already rather complex picture of delocalization. In order to more or less correctly estimate the contribution of certain bonds to the observed distribution of fundamental vibrational frequencies and the distribution of band intensities, it would be necessary to analyze the normal vibrations of a large number of isotopomers containing isotopes H, D, N^14^, N^15^, O^16^, and O^18^ in various combinations. Due to the lack of information of this kind, we had to confine ourselves to general information on frequencies in individual areas, which is given in the monograph [45]: 3100–3000 cm^−1^ frequency range of stretching vibrations of aromatic bonds CH; 1600–1500 cm^−1^ frequency range of stretching vibrations of the CC with the contribution of bending vibrations of the CCC and CCH of the phenyl ring; 1500–1450 cm^−1^ the frequency range of deformation vibrations of the CCH; 1300–1000 cm^−1^ frequency range of bending vibrations of the CCH with the participation of bond angles and bonds of the CC ring; 900–675cm^−1^ frequency range of out-of-plane vibrations of CH bonds with the participation of out-of-plane vibrations of the ring; 580–450 cm^−1^ the frequency range of out-of-plane vibrations of the CH bonds and the ring. The mixing of the stretching vibrations of the CC and CN bonds in the aromatic ring, which is related to the pyrazine molecule, excludes any possibility of a separate assignment of frequencies for these bonds.

The PPQ film investigated by IR spectroscopy is anhydrous, as evidenced by the spectrum free of any absorption bands in the 3100–4000 cm^−1^ region. Consider the overview IR absorption spectrum of the SPPQ 5.5% film (Figure 8). It is noteworthy that, upon sulfonation of PPQ, the vibrational frequencies, and intensities of most of the bands, as a rule, do not change. This fact indicates that there is no violation of the conjugation of bonds in the heterocycle. Sulfonation leads to the appearance of an intense band at 1192 cm^−1^. High intensity and noticeable broadening of the contour, not to mention the value of the vibration frequency, are quite characteristic of the band of the antisymmetric stretching vibration ν_as_(SO_3_) of the sulfogroup. A narrow band of average intensity at 1033 cm^−1^ can be attributed in the first approximation to the symmetric stretching vibration of the ν_s_(SO_3_) sulfogroup. There is a band of in-plane deformation vibrations of CCC and a band of in-plane bending vibrations of CCC with the participation of bond angles CCC are located nearby, as well as stretching vibrations of the ring. The maximum of the first of them is at 1019 cm^−1^, and the maximum of the second is at 1066 cm^−1^. Small differences in frequencies create favorable conditions for mixing all these vibrations.

The bands of deformational antisymmetric and symmetric vibrations of the sulfogroup are usually of low intensity. In the case of interest to us, these vibrations do not lead to the appearance of separate bands, partly for this reason and also due to mixing with out-of-plane vibrations of the CH bonds and the ring. However, the obvious enhancement of the band at a frequency of 616 cm^−1^, which is observed on going from PPQ to SPPQ, can be caused by a noticeable contribution of the δ_as_(SO_3_) vibration.

For the band of average intensity at 1699, 1715 cm^−1^ refers to stretching vibrations of the C=O bond. It is absent in the spectrum of the PPQ film, which indicates the appearance of an impurity of one of the initial two reagents, namely tetraketone BFGB, in the sample under study upon sulfonation. In contrast, TADO tetramine bands are not detectable. It was found that the (CO) band of tetraketone disappears after the treatment of SPPQ with an aqueous solution of NaOH.

The band at 3059 cm^−1^ (shoulder 3016 cm^−1^) has an average intensity and a narrow profile in the IR spectrum of the PPQ film. On the side of low frequencies from it, there are two much less intense and, at the same time, very narrow bands. The maximum of one of them is at 2917 cm^−1^, and the other at 2849 cm^−1^. It is obvious that all three bands are related to the stretching vibrations of CH bonds. A similar band with a narrow profile and a maximum at 3060 cm^–1^ is also present in the IR absorption spectrum of the SPPQ film. At the same time, sulfonation of the PPQ film leads to the appearance of an intense band with a very wide and asymmetric band contour, the maximum of which is at 3458 cm^−1^. Its origin is associated with the presence of proton hydrates in the sample. The low-frequency wing of the band ν(OH) has a continual character; that is, it is characterized by a rather high intensity and, at the same time, a significant length, which is usually observed during the hydration of a proton in various media. One of the three above-mentioned bands ν(CH) is satisfactorily resolved at 3065 cm^−1^ against the background of the continuum due to its relatively narrow profile and optimal intensity. Whereas narrower bands ν(CH), which should have been at 2849 and 2917 cm^−1^, on the contrary, are not resolved due to their low intensity under conditions of screening by intense continual absorption. Attention should be paid to the fact that the contour of the strip ν(OH) changes its shape during the dehydration of the PPQ film kept in a humid atmosphere. Figure 9 gives an idea of the nature of the change due to the removal of moisture under vacuum at room temperature. It can be seen that when the degree of hydration is high (upper curve), the maximum of the wide asymmetric band ν(OH) approaches the maximum at 3420–3450 cm^–1^, which is essentially the same as the broad band of stretching vibrations of the liquid water molecule. The stretching vibration frequency for the film is 3458 cm^–1^, and its value practically does not change at some intermediate stages of sample dehydration.

Spectroscopic information allows us to conclude that the SPPQ film adsorbs water in a sufficiently large amount, primarily due to its proton and sulfogroup binding. Moreover, the possibility of additional (physical) adsorption of water molecules in capillaries is not excluded. In the case of physical adsorption, tetrahedrally directed hydrogen bonds, similar to those in liquid water, can arise between H_2_O molecules. It is known that the ν(OH) band of liquid water additionally exhibits a shoulder at 3225 cm^−1^, the origin of which is explained by the superposition of the overtone 2ν_2_, enhanced by the Fermi resonance [46]. The same shoulder is observed at 3203–3254 cm^−1^ on the contour of the intense band ν(OH) of a film with a high content of adsorbed H_2_O molecules (Figure 9).

In order to distinguish between the spectra of physically adsorbed water molecules and the spectra, in a favorable case, the OH bending vibration bands could be used. However, one of these bands is screened in the film by a ν(CO) band at 1713 cm^−1^, and the other is screened by a band of stretching vibrations of the ring at 1615 cm^−1^.

As shown in Figure 9, holding the film under a vacuum is accompanied by a noticeable drop in the intensity of the maximum ν(OH) at a frequency of 3458 cm^−1^. It should be noted that the shoulder at 3203–3254 cm^−1^, contrary to expectations, does not disappear when the intensity of the maximum at 3458 cm^−1^ becomes small due to the loss of a certain amount of water molecules adsorbed by the film. The reason is the appearance of a new maximum at a frequency of 3203 cm^−1^ with a moderately wide profile. At intermediate stages of dehydration in the IR spectrum of the SPPQ film, two maxima of comparable intensity and width with a slight frequency drift are resolved. The position of one of them corresponds to a frequency of ≤3458 cm^−1^ and the other to a frequency of ≤3203 cm^−1^. Judging by the found value of the frequency, the new maximum should be attributed to the stretching vibrations of the OH cation H_5_O_2_^+^. Further exposure of the film under a vacuum ultimately leads to the disappearance of this maximum as well as the previous one. After several hours of exposure, the spectrum as a whole ceases to change. The minimum moisture content of the SPPQ contributes to the formation of a very wide band contour with a maximum of 2850 cm^−1^ and a shoulder at 2540–2630 cm^−1^ on the low-frequency wing. Its high-frequency wing has shoulders left after the disappearance of the previous two maxima. The intensity and outline of this change do not undergo further exposure to the SPPQ film under a vacuum. The spontaneous stopping of the dehydration process at room temperature while the holding of the film under vacuum continues indicates a strong binding of residual water in the film. This result, combined with a very large decrease in the OH stretching vibration frequency, indicates the formation of a lower proton hydrate. Therefore, the only ν(OH) band with a maximum at 2850 cm^–1^ observed upon prolonged exposure of the film under vacuum must belong to the stretching vibration of the H_3_O^+^ cation. Differentiation between the ν(OH) bands belonging to different cations becomes possible due to the fact that the H_3_O^+^ cation forms strong hydrogen bonds with the oxygen atoms of the sulfogroup and the H_5_O_2_^+^ cation forms noticeably weaker hydrogen bonds. The weakest hydrogen bonds are formed between the higher hydrates of the proton and the oxygen atoms of the sulfogroup.

Studies of the physical properties of the polymer showed that the mechanical strength of the SPPQ film is low, but the sample continues to remain quite elastic both at the maximum saturation with water molecules and at the lowest possible (residual) content. Extremely dehydrated SPPQ film is very hygroscopic. When exposed to air, it quickly absorbs moisture from the environment. The band at 2850 cm^−1^ then disappears, giving way to a continuum, and instead of it, already known maxima ν(OH) appear—first belonging to the H_5_O_2_^+^ cation and then to higher proton hydrates; the result depends on the ambient humidity. Different moisture content and the formation of different types of hydrated protons are reflected, as follows from Figure 9, on the parameters of the band ν(OH) to a very large extent. On the contrary, the vibrational spectrum of aromatic cycles in the composition of the polymer does not undergo noticeable changes upon saturation of the film with water molecules and upon their removal.

Figure 9b demonstrates the dependence of the band profile ν(OD) on the content of D_2_O molecules in the film SPPQ. The shape of the contour changes with the duration of exposure to the deuterated film under a vacuum, similar to the change in the shape of the contour ν(OH) in Figure 9. Stretching vibrations of physically adsorbed D_2_O molecules and higher hydrates D^+^ caused the appearance of an intense asymmetric band ν(OD) at 2526 cm^−1^ in SPPQ films saturated with heavy water. The stretching vibrations of the D_5_O_2_^+^ cation cause the appearance of a moderately wide band ν(OD) at 2389 cm^−1^, and the stretching vibrations of the D_3_O^+^ cation give rise to a very wide band ν(OD) at 2263 cm^−1^. The first appears approximately 10 min after the beginning of exposure of such films under vacuum, and the second, after 2 h or more. The frequency range 1200–1270 cm^−1^, which is most characteristic of bending vibrations OD of physically adsorbed D_2_O molecules and D^+^ hydrates, is closed by intense vibration bands ν_as_(SO_3_), δ(CCH), and ν(CC).

### 3.6. Quantum Chemical Simulation of Structure of SPPQ

At the calculations, two variants of the arrangement of benzene rings in the initial polymer were considered (Figure 10, structures **1** and **2**).

The structure with benzene rings in the transposition turned out to be energetically more favorable; the difference between the configurations was small (~0.06 eV). Next, various options for the addition of the sulfogroup to the main chain of the polymer were considered (Figure 10, structures **3**–**7**). In this case, configuration **3** with a sulfogroup on the C-C bond near the bridging oxygen atom was most favorable, and the difference with other structures turned out to be quite noticeable (0.2–0.4 eV). From configuration **1** and a unit with two sulfogroups **8**, double units of the initial polymer were formed (Figure 11, **1**–**3**).

### 3.7. Impedance Spectroscopy

The conductivity of the initial PPQ in the form of a powder and a film does not exceed 10^−8^ S/cm, even at a humidity of 95 rel. %. The investigation of the influence of ambient humidity on the conductivity of SPPQ in tablet form from the powders of the initial SPPQ showed that the proton conductivity with increasing humidity increases to 10^−4^ S/cm for SPPQ 5.5% (Figure 12). The activation energy of conductivity at a humidity of 42 rel. % is 0.4 eV and decreases to 0.2 eV at 95 rel. %.

Long-term measurements (over 7 years) of proton conductivity were carried out on SPPQ film samples with various degrees of sulfonation (sulfur content 2.6, 4.2, 5.5, and 7%). The proton conductivity of SPPQ 7% after long-term measurements reached8× 10^−4^ S/cm at 98 rel. %. As can be seen from Figure 13, SPPQ, with a sulfur content from 2.6 to 5.5, hasfairly close conductivity values.

Earlier in our work [27], we tested SSPQ as a proton-conducting membrane in a membrane electrode block of a fuel cell. The results of the electrochemical measurements of the power characteristics of the membrane electrode block with the membranes based on sulfonated PPQs with sulfur contents of 6.0 and 8.0% are presented in Figure 14 [27]. The characteristics of the fuel cells, to a significant extent, depend on the degree of sulfonation of PPQs. The maximum power developed by the membrane electrode block with a membrane based on sulfonated PPQs with a sulfur content of 8.0% is ~80 mW/cm^2^ (Figure 14) [27]. The power obtained on a Nafion NRE-212 membrane in the case of the use of electrodes prepared by the same method under similar measurement conditions was 120 mW/cm^2^.

## 4. Conclusions

It was shown in this work that sulfonated polyphenylquinoxalines have high thermal stability and proton conductivity at ambient humidity of 98 rel. % reaches values of 10^−6^–10^−3^ S/cm, depending on the degree of sulfonation. The relative stable configurations with different positions of benzene rings and sulfogroups with the help of quantum chemical modeling were considered. Long-term measurements of proton conductivity were carried out. These objects can be quite promising if it is possible to improve the mechanical properties and increase the conductivity in a wider range of humidity.

## Data Availability

Not applicable.

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
