# Peer review of "Structure, Thermal Properties and Proton Conductivity of the Sulfonated Polyphenylquinoxalines"

_membranes, 2022, doi:10.3390/membranes12111095_

Round 1
Reviewer 1 Report
Overall comments: In this work, the authors synthesized sulfonated polyphenylquinoxalines and investigated the structure, thermal stability, water uptake, and proton conductivity of the sulfonated polyphenylquinoxalines with different sulfur content. Quantum chemical modeling was utilized to identify the stable configuration of the SPPQ. However, the novelty of the study is not sufficient since there has been previous literature on the preparation and characterizations of sulfonated polyphenylquinoxalines. In addition, the manuscript lacks significant experimental results and the reported data has flaws and misinterpretations. Discussion of the data in the manuscript (thermal properties, water content, conductivity, etc) is not sufficient. Hence, the manuscript is not suitable for publication in Membranes.
Major comments:
1. line 31 – 34: Literature review on previous investigations of sulfonated aromatic condensation polymers needs to be added to the manuscript. Why sulfonated aromatic condensation polymers attract significant interest among different types of polymers?
2. The novelty of the study is not clearly discussed in the manuscript. The authors introduced previous studies on PPQs as proton exchange membranes, including acid doping PPQs, SPPQs. Studies on the dependence of proton conductivity on the degree of sulfonation, temperature, humidity, and anion type of the dopants were referenced in the introduction section of the manuscripts. The influence of the entanglement density on the deformation of polymers is also included in the introduction. The results in this study have mostly been covered in previous studies.
3. Line 159, water sorption properties: The author mentioned that the equilibrium time for the water content of different samples varied from one week to a month. The time for equilibrium needs to be provided for each polymer. Is there any trend in the equilibrium time vs sulfonation level?
4. Line 171 – 173: The authors determined the molecular weight of PPQ and SPPQ using GPC that was pre-calibrated with polystyrene standards. The shape of the GPC curves in Figure 4 is not smooth. The authors did not explain the uncommon shape of the GPC results. Ionic polymers (SPPQ) often have electrostatic repulsion and chain aggregation, which could result in inaccurate elution time and molecular weight results. Therefore, the current GPC results are not reliable for molecular weight determination. The authors need to minimize the ionic polymer chain aggregation in GPC or use a different characterization method to determine the molecular weight of the polymers.
5. Line 185 – 188: Why do the authors use different electrodes for testing electrolytes made from powder (Pt as the electrode) and membrane (C as the electrode)?
6. A lot of the data in the figures and tables used “,” instead of “.” as decimal points. This is not correct and very misleading for the interpretation of data. Please correct accordingly.
7. Table 1: There is no discussion of the data in Table 1 in the manuscript. What are the water content data for temperatures higher than 100 °C? Do the authors attribute all of the weight loss of SPPQ below 300 °C to water? The sulfonate group would start to decompose above 250 °C and proof that the weight loss is not solvent or other impurities is needed to attribute the weight loss to water. The authors need to add NMR figures to confirm the structure and purity of PPQ and SPPQs.
8. Figure 3: Why is the water sorption data for SPPQ films so low? Literature on SPPQs showed significant water uptake (10 – 30% at 20°C in liquid water, Ref [25]; 26 – 45% at 30 °C, Liang et al, Membranes 2022, 12(10), 952). However, the water sorption is only 0.5% for SPPQ with 7% of sulfur content at 98% relative humidity. Why is the huge difference compared to the literature and low water sorption for these polymers?
9. Figure 5: How were Figure 5c and 5d determined to be the sulfonation regions of SPPQs? Have the authors used energy dispersive X-ray spectroscopy (EDS) to analyze sulfur distribution?
10. Figure 6: Characteristic of the bands need to be labeled in Figure 6. It will be helpful to interpret the figure with the labeled band.
11. Figure 10 and Figure 11: The discussion on proton conductivity data is not sufficient. Why are multiple symbols used for the conductivity of PPQ? How does the conductivity of these polymers compare to the benchmark Nafion? Is there any dependence of proton conductivity on the degree of sulfonation? It seems that conductivity is similar for SPPQs with a sulfur content of less than 7%.
Typos and grammar comments:
1. There are multiple typos and grammar mistakes in the manuscripts. The authors need to carefully recheck the entire manuscript and fix all errors to improve the quality of the manuscript. For examples:
Line 60: “The authors of [13] were sulfonated the PPQ film…” need to change to “The authors of [13] sulfonated the PPQ film”
Line 151: “…the sulfur content in the SPPQ samples 2.6, 4.2…” need to change to “…the sulfur content in the SPPQ samples is 2.6, 4.2…”
Line 153: “Thermal stability of PPQ…” need to change to “Thermal stability. The thermal properties of PPQ…” to keep consistent with other subheadings.
2. In the materials and methods sections, the grade of the chemicals used in the study needs to be added. What is the equivalent degree of sulfonation (%) for SPPQ with different sulfur content (0, 2.6, 4.2, 5.5, 7%)?
3. line 204 – 206: The template of the journal was still left in the manuscript. Please delete.
4. line 208: There is no table number in the table legend. It should be “Table 1.”
Reviewer 2 Report
The manuscript reported the preparation, structure, thermal properties, and proton conductivity of sulfonated polyphenylquinoxalines. It is found that sulfonated polyphenylquinoxalines contain water and are thermally stable up to 250 °C. The proton conductivity at an ambient humidity of 98 rel. % reaches values 10-6-10-3 S/cm depending on the degree of sulfonation. Even after long-term storage in air, samples with a high sulfur content of 7% at 98% air humidity maintain a conductivity of 8*10-4 S/cm.
The content of this manuscript meets the reading interests of the readers of the journal. However, there are certain English spelling and grammar issues, and also the discussion and explanation should be further improved. I suggest giving a minor revision and the authors need to clarify some issues or supply some more experimental data to enrich the content.
1. For grammar issues, it is suggested that the author double-check the small grammar errors in the full text, especially the lack of and redundant use of definite articles.
2. For the Keywords, ‘quantum chemical modeling’, ‘thermal stability’, and ‘degree of sulfonation’ should be added in order to attract a broader readership.
3. Page 1, ‘The synthesis of new polymer proton-conducting electrolytes [1], which are one of the main components of fuel cells [2, 3], attracts considerable attention of researchers [4].’ What is the problem or drawback of the current commercial PEMs? It should be explained further. For example, Nafion series membranes are costly due to the unique perfluorinated sulfonic acid structures, and PEMs with higher conductivity is required, and so on (10.1080/15583724.2019.1641514; 10.1016/j.electacta.2019.03.056). And for ‘different aromatic condensation polymers [5, 6], poly(arylene oxides) [7], polybenzimidazole [8]’, PEEK should also be mentioned since it is widely investigated as SPEEK as well.
In addition, ‘due to solubility in organic solvents and the ease of polymer-analogous transformations - in particular, sulfonation: thermoreactive polymers [12], different methods of sulfonation [13-15], polymers in salts form [16], modified polymers [17].’ This sentence does not read coherently. There are many fragmented languages and the logic is not particularly clear. It is recommended to rewrite this part.
4. Page 2, ‘The disadvantage of this type of membrane is the solubility of the original polymer in adsorbed phosphoric acid.’ It is not so clear that the disadvantage is the solubility is too high or too low. For the membrane doping with phosphoric acid, it is important to have a suitable acid-doping level, so the acid uptake/absorbance for the membrane is important. Moreover, the membrane should be chemically stable with the doping acid. What is the relation between polymer solubility and acid uptake/doping level? Do the authors indicate that the polymer membrane can be easily soluble in phosphoric acid so that the durability for doping is bad?
From References 23-27, the descriptions of different membranes with various conductivities are very mixed-up, and it is not very clear to understand the differences between them, or which way can obtain membranes with optimal properties.
‘Each state has its own colour: pale yellow like that of PPC’. What is PPC? Also, in the manuscript, there is also SPPC. I consider only PPQ and SPPQ should be correct, and the authors should double-check these issues.
5. Page 5, ‘This section may be divided by subheadings. It should provide a concise and precise description of the experimental results, their interpretation, and the experimental conclusions that can be drawn’ should be deleted.
For the TGA result, it is not so clear. It is better to add the first derivative of the wt% in the temperature range ( see Figure 1 of 10.1016/j.ssi.2018.01.038).
6. Page 7, ‘Unfortunately, one of the drawbacks that hinder the practical use of the obtained SPPQ is the brittleness of the films under study.’ What can be done to further solve the brittle properties of the membranes? Inorganic-organic hybrid or polymer blendings? Some possible solutions should be given after showing the SEM.
7. Why is there no chemical stability test and fuel cell polarization curve test in the full text? At the same time, the SPPQ in this paper needs to be compared with the characteristics of commercial membrane Nafion. Otherwise, how can we prove that SPPQ material is expected to replace commercial membrane Nafion?
Reviewer 3 Report
The paper investigated the structure, thermal properties and proton conductivity with different degree of sulfonation of the polyphenyl quinoxalines (SPPQ). The paper will be accepted after the two questions should be answered.
1. The proton conductivity of SPPQ film should be compared with other membranes.
2. The paper should give the performance of fuel cells using the SPPQ film, which can really reflect the electrochemical performance of the SPPQ film.
Round 2
Reviewer 1 Report
The manuscript is acceptable to be published in Membranes.